# Training Scientific Communication Skills on Medical Imaging within the Virtual World Second Life: Perception of Biomedical Engineering Students

**DOI:** 10.3390/ijerph20031697

**Published:** 2023-01-17

**Authors:** Andrés Lozano-Durán, Teodoro Rudolphi-Solero, Enrique Nava-Baro, Miguel José Ruiz-Gómez, Francisco Sendra-Portero

**Affiliations:** 1Communications Engineering Department, Higher Technical School of Computer Engineering, University of Malaga, 29071 Malaga, Spain; 2Department of Radiology and Physical Medicine, School of Medicine, University of Malaga, 29071 Malaga, Spain

**Keywords:** online learning, virtual worlds, biomedical engineering, undergraduate education, radiology, medical imaging

## Abstract

Second Life is a multi-user virtual world platform which enables online learning through immersive activities. This study evaluates the perception of third-year biomedical engineering students about learning activities complementary to a biomedical imaging course carried out within Second Life and focused on training in the public presentation of scientific content to their peers. Between 2015 and 2017, students gave oral presentations on medical imaging topics selected from the proposals of their classmates. Participants were invited to complete an evaluation questionnaire. In the three years of the study, 133 students enrolled in the course (48, 46, and 39 consecutively), and 97 of them delivered the questionnaire (48%, 83%, and 92%, consecutively). Attendance at the sessions ranged between 88% and 44%. The students positively value the experiences, especially the teacher, the educational content, and the virtual island environment, with mean scores greater than or equal to 8.4, 7.7, and 7.7, respectively, on a 1–10-point scale. Overall, they valued Second Life as an attractive and suitable environment for their training in science communication skills, in which they gain self-confidence and are less afraid of speaking in public. Second Life enables students to present scientific content effectively to their peers, receiving hands-on training in the tasks of collecting, organizing, and presenting data, with the benefits of remote access, collaborative work, and social interaction.

## 1. Introduction

### 1.1. Virtual Worlds in Health-Related Undergraduate Education

The concept of virtual worlds, along with others such as virtual reality, mirror worlds and augmented reality, integrates the concept of the metaverse [1], a word composed of “meta”, which means transcendence and virtuality, and “universe”, which means world and universe, a virtual reality that exists beyond reality, referring to the digitized Earth as a new world expressed through digital media and the Internet. Three-dimensional (3D) virtual worlds are computer-generated settings that represent 3D scenes on the computer screen with a representation of the users called an avatar [2]. Virtual worlds are multi-user platforms that support a set of human activities that improve the ways of learning by allowing file sharing, immersive lectures, and simulations [3].

Second Life (Linden Lab, San Francisco, CA, USA) has been one of the most used virtual worlds in higher education since its release in 2003 [2,4,5,6,7]. For several years, healthcare educational experiences have been carried out on this platform, which allows users to interact with each other, communicate by voice or chat, and send deferred messages through notecards [8]. Second Life grants the possibility of communicating over long distances with a high sense of presence. A positive impact of Second Life in higher education has been described [5] as: (a) producing a great immersive feeling of “being there” in the user, (b) allowing a great variety of interactions with the elements of the virtual world and other users, (c) allowing visualizing and contextualizing objects and images, (d) enabling the exhibition of content and culture, (e) the ability to work individually or collectively, (f) enabling the creation of simulation environments, (g) providing a sense of community to the user, and (h) allowing the creation of content. There are multiple experiences regarding the effectiveness of Second Life in the learning of health professions students [2,3]. Specifically, in the field of radiology, various experiences have been carried out with medical students, demonstrating that radiological images can be used in educational presentations with adequate quality for learning and that the usual classroom activities (lectures, courses, workshops, etc.) can be reproduced in 3D environments with very good evaluation and acceptance by students [9,10] and with an impact on learning equal to that of the conventional environment [11].

### 1.2. Radiology Teaching to Biomedical Engineering Students

Undergraduate biomedical engineering education should contemplate training in medical imaging as basic elements to understand the fundamentals and technology of image acquisition. The teaching of radiology and imaging methods in clinical practice must be one of the basic pillars of undergraduate education for biomedical engineers [12,13]. Radiology is now an essential part of the medical diagnostic process and has undergone a tremendous transformation during recent years mainly due to the technological and technical development in this field [14]. Medical imaging acquisition and processing has evolved as an interdisciplinary subject in which education involves mathematical, physics, engineering, and medical knowledge. These techniques have become crucial for diagnosis, clinical applications, and research activities, thus having a broad impact on the curricula of biomedical engineers [15].

The education of today’s engineering students requires incorporating immersive online techniques that energize their training and engage them with their undergraduate learning. Some multidisciplinary experiences involving clinical simulations that showed positive feedback and granted a holistic view of the healthcare process have been developed with biomedical engineering students [16]. Additionally, the need of a multidisciplinary space where biomedical engineering students can meet other healthcare students to resolve situations altogether has been described [17]. These virtual simulation experiences could provide students an interesting approach to everyday clinical situations [18].

### 1.3. Background

The University of Malaga began to offer the biomedical engineering degree in 2011. Since 2013, a four-month course called Biomedical Imaging is taught in the third year, the teaching of which is shared between the Department of Communications Engineering and the Department of Radiology and Physical Medicine. This course is an introduction to radiology and clinical imaging and provides students with a broad overview of the main engineering, mathematical, physical, computational, and diagnostic concepts behind the biomedical imaging modalities. Teaching biomedical imaging is scheduled in the second half of the four-year undergraduate curriculum, when students have acquired enough knowledge of other disciplines to understand the fundamentals and the usefulness of these imaging techniques.

In 2011, a space was developed in Second Life to explore immersive online radiology teaching. Since then, educational activities have been carried out in this virtual location, with undergraduate and graduate students currently exceeding 3000 users. This study describes the first learning activities in Second Life with biomedical engineering students, focused on training oral communication skills in public.

### 1.4. Objectives

This study evaluates the perception of third-year biomedical engineering students about learning activities complementary to a biomedical imaging course carried out within the Second Life virtual world and focused on training in the public presentation of scientific content to their peers.

## 2. Materials and Methods

### 2.1. The Virtual Environment: The Medical Master Island

The Medical Master Island is a 3D virtual setting created in Second Life in 2011 to offer innovative educational experiences in healthcare learning, mainly about radiology. This environment mimics a university campus on an island, with several buildings arranged around a central esplanade, connected to each other via walkways [9]. In addition, to boosting the playground of the platform, the island houses other settings such as caves, palaces, submerged cellars, etc. A floating auditorium was created 120 m above the ground to carry out the activities with biomedical engineering students (Figure 1a).

In Second Life, the elements displayed to the user are made by primary objects called prims, which can reproduce web pages on any of their faces (Figure 1b). This resource can be used to build a flat panel with a web page reproduced on its main face so that a slideshow with educational content can be easily created. These slide shows were created from PowerPoint presentations, saved as JPEG files (one file for each slide), and then linked to simple web pages with two back and forward buttons (Figure 1c,d).

### 2.2. The Course Biomedical Imaging

Biomedical Imaging is a third-year course that addresses important aspects of the formation, representation, treatment and processing, recognition, and interpretation of the digital image in a biomedical context. The syllabus is divided into four modules:Biomedical imaging and modalities: introduction to biomedical imaging, in which the main types of medical imaging modalities are described, such as digital radiography, computed tomography, ultrasound imaging, magnetic resonance, and nuclear medicine.Compression, rendering and enhancement of biomedical digital images: The main image formats are studied, including the DICOM format. Image enhancement techniques and segmentation of regions of interest are also covered.Biomedical image processing and analysis: The main techniques of mathematical morphology, the use of image descriptors for their characterization, and the reconstruction of images from projections are studied.Laboratory activities: This subject has a strong practical component, in which the student can put into practice all the theoretical knowledge acquired by carrying out activities such as the handling of biomedical images in DICOM format, digital image filters, and the segmentation of biomedical images, including extraction of objects of interest.

The activities that make up this study were carried out in Second Life throughout three courses: first, a pilot experience was carried out with students as presenters of educational content, and then two experiences were carried out to train students in delivering oral presentations to their peers.

### 2.3. Pilot Experience with Students as Presenters of Educational Content

It was carried out between 20 October and 24 November 2015 as a voluntary activity of the course. Students prepared topics and presented them orally to the teacher and the rest of the class, individually (for 20 min) or in groups of three students (for 30–40 min). Second Life viewers were installed in the computer school, on the computers in the library, and in the lab. The activity was organized into five 2-h sessions:1.Training and a visit to the island. Radiobiology and Radiological Protection Seminar, given by the professor. Discussion and proposal of topics for the next few days.2.Oral presentations by students3.Oral presentations by students4.Oral presentations by students5.Oral presentations by students

At the end of Session 5, attendants were asked to evaluate the presentations given by their classmates, delivering a notecard indicating the best and the second- and third-place ones, following their own criteria. Afterwards, a score system was established, giving 1 point, 0.5 points, or 0.3 points, respectively, each time a presentation obtained the first, second, or third position. The sum of scores was normalized to 10 points.

Finally, the students were asked to fill out a questionnaire consisting of 22 statements to be answered with a Likert scale from 1 (completely disagree) to 5 (completely agree); 9 aspects of the experience to score from 1 to 10 points; and a space to add open comments. This questionnaire, used in other published studies on teaching experiences with medical students [9,10,11,19,20], included minimal modifications adapting the statements to the present study. The questionnaire showed good reliability for the statements related to the teaching activity and the global evaluation of the project (Cronbach’s alpha ≥ 0.84) and acceptable reliability in the statements about the students’ experience in Second Life (Cronbach’s alpha ≥ 0.70). Kendall’s Tau-b analysis showed a positive correlation between all paired statement comparisons related to global assessment (Tau-b > 0.22, *p* < 0.01) and students’ perception of the experience (Tau-b > 0.15, *p* < 0.05) [19,20]

### 2.4. Training of Students in Delivering Oral Presentations

The same experience was carried out as a mandatory activity of the course from 13 October to 16 November 2016 and from 18 October to 13 December 2017. Attendance accounted for 10% of the final grade in this activity, but the development of the presentations had no impact on the grade for the subject. These aspects were duly explained to the students. The activity was developed in five 2-hour sessions:1.Reception and training in Second Life. Explanation of the activities to be carried out.2.Presentation of web pages. To train the oral presentation in Second Life, the students had to choose a web page of interest for the course and present it to their classmates for 5 min.3.Oral presentations by students4.Oral presentations by students5.Oral presentations by students

The students evaluated their classmates’ presentations by scoring 1–10 points for the content, quality and understandability of the presentation, and the interest in the topic. At the end of the experience, they completed an evaluation questionnaire identical to the previous experience in 2015, including an additional statement about conducting a first session presenting web pages, to be answered with a Likert 1–5 scale. The questionnaires used in this study and the open comments given by respondents, translated into English, can be read in detail in Appendix A, respectively.

### 2.5. Data Analysis

Excel 2019 (Microsoft, Redmond, WA, USA) was used to organize data and de SPSS statistical package v24 (IBM Corporation, Armonk, NY, USA) and R Studio v1.4 (R Studio, Boston, MA, USA) were used for the statistical analysis. The number of students that assisted to each seminar and the number of questionnaires delivered are quantitative variables. The numerical answers of the perception questionnaire are treated as ordinal variables and are presented un terms of mean ± standard deviation (±SD). Mann–Whitney U test was used to assess differences in the five-point Likert scale and 1–10 point global evaluation between different groups of students and between Adobe Connect and Second Life items in the 2014 cohort. Statistical significance was accepted when a probability of error *p* < 0.05 was obtained.

Open comments were analyzed by theme analysis using systematic collaborative coding [21]. A first coding was carried out independently by two of the authors, and later the codes were refined and definitively assigned in group consensus meetings.

### 2.6. Ethical Approval

All the data from the questionnaires were anonymized, following current legislation on the protection of personal data and the guarantee of digital rights. This study takes part of the project entitled “Teaching experiences on medical imaging with biomedical engineering students in the virtual world Second Life”, which received the approval of the ethics committee for experimentation at the University of Malaga (decision number 158-2021-H).

## 3. Results

### 3.1. Pilot Experience with Students as Presenters of Educational Content

In 2015, there were 48 students enrolled in the course, and 38 (79.2%) attended at least three of the five sessions. Five students (10.4%) did not participate in any session. The attendance percentage decreased from 75–88% in the first 3 days to 56–44% in the last 2 days (Figure 2).

At the end of the activity, 23 students (47.9%) delivered the perception questionnaire about the experience. All of them attended at least 3 of the 5 sessions, except for 2 students who attended 2 and none. Only 5 students (21.7%) were familiar with Second Life before this experience, and the rest did not know it. Figure 3 and Figure 4 show the mean of the quantitative data from the questionnaire compared with the 2016 and 2017 cohorts. The best valued statements on the Likert scale from 1 (completely disagree) to 5 (completely agree) were: the teacher’s intervention was adequate (4.6 ± 0.6); the colleagues who presented the topics did very well (4.2 ± 0.7); the creation and management of the avatar was easy (4.2 ± 0.9); the floating auditorium was adequate to hold the sessions (4.2 ± 0.9); and the island environment seemed attractive (4.1 ± 1.0). The students did not find the content difficult or the number of sessions excessive.

All the evaluations of 1–10 points had an average higher than 7.3 points (Figure 3), highlighting the teacher (9.0 ± 1.2), the environment of the island (7.9 ± 1.4), the educational content (7.7 ± 1.7), and the sessions in Second Life (7.8 ± 1.7). Some significant differences were found compared to the evaluations carried out the previous year. In 2015, students scored higher on the interaction with their classmates (7.9 ± 1.3 versus 6.5 ± 1.4; *p* = 0.001), the island environment (7.9 ± 1.4 versus 6.7 ± 1.9; *p* = 0.014), and the sessions within Second Life (7.8 ± 1.7 versus 6.7 ± 1.4; *p* = 0.024).

There were only 5 open comments in the 23 questionnaires delivered (21.7%), including 10 positive and 3 negative comments and 1 suggestion proposing to shorten the duration of the sessions a bit (Table 1). The positive comments focused primarily on the educational, innovative, and fun aspects of the experience. Among the negatives, one referred to technical problems and two to the excessive time length of the sessions or the entire experience.

Table 2 shows the score given by the students at the end of Session 5 with a system based on individually classifying the three best presentations. This system evaluates the presentation as a whole and could eventually leave some presentations without being evaluated. For this reason, it was decided to change the peer evaluation system in the following years, assessing specific aspects such as the content, interest, quality, and understandability of all the presentations.

### 3.2. Training of Students in Delivering Oral Presentations

In 2016, 46 students enrolled in the course and 39 (84.8%) attended at least 3 of the 5 sessions. Six students (13.0%) did not participate in any session. In 2017, 39 students enrolled in the course and 35 (89.7%) attended at least 3 of the 5 sessions. Everyone participated in at least one session. Figure 2 shows the percentages of attendance to each session. In 2016, 12 topics were selected from those proposed by the students. In 2017 the same themes were used plus two additional ones. Table 3 presents the title of the topics and the peer evaluation in terms of mean ± standard deviation. In general, the scores given in 2017 were lower than in 2016. In both years, the assessment of the interest in the topics stood out.

The perception questionnaire was filled out by 38 students in 2016 and 36 students in 2017. Few students were familiar with Second Life before this experience, i.e., four (10.5%) in 2016 and only one (2.8%) in 2017. In both cohorts, students showed greater agreement that teacher intervention, topic selection, and auditorium were adequate, the island environment was attractive, and the tasks of creating and managing avatars were easy, with a mean equal to or greater than 4 on a Likert scale of 1–5 (Figure 3). Unlike the 2015 experience, a higher proportion of students found the number of sessions excessive (*p* = 0.015 and 0.017).

Regarding the score of 1–10 points, the best valued was the teacher, followed by the educational content, and the environment of the island (Figure 3). The 2017 group gave significantly lower scores in some aspects, such as the overall experience, the interaction with their peers, or the sessions in Second Life (Figure 3), and they also showed less agreement in considering the content as appropriate for their education, finding interest in the initiative, or in being willing to repeat the experience (Figure 4).

The students included open comments in 35 questionnaires—20 in 2016 (52.6%) and 15 in 2017 (41.7%), with 62 positive, 50 negative, and 10 suggestion codes (Table 1). They highlighted in their positive comments that the experience was interesting (19 comments), educational (17 comments), and innovative (12 comments). They also expressed the advantages of ubiquity or fun and expressed their willingness to participate in other similar experiences or their gratitude for the work done. They highlighted as negative comments the technical limitations to follow the sessions in Second Life properly (15 comments), the excessive time of the sessions or the experience as a whole (10 comments), schedule conflicts (7 comments), or the preference for face-to-face activities (7 comments). Five students commented on their dissatisfaction with the peer review, finding it subjective. Two students complained about the need for Second Life to be accessible from computers located in the school. Ten students provided interesting suggestions on modifications of the schedule, the duration of the sessions, the selection of topics, restructuring of the exhibition groups, reinforcement of the training in communication by microphone and audio, or inclusion of the teacher’s assessment in addition to that of the peers.

## 4. Discussion

Biomedical imaging is an important area for the biomedical engineering profession, and strong undergraduate imaging programs are critical to developing the essential human infrastructure necessary to support this area [13]. Innovative technology efforts are needed to engage current biomedical engineering students in this area. Three-dimensional virtual worlds such as Second Life constitute a promising innovation in the field of information and communication technologies [22], with unique characteristics that can have a positive impact on teaching and learning, such as immersion, sense of presence, visualization, contextualization, simulation of content, and individual or collective interactions [5]. In this study, the teaching activities of a biomedical imaging course were implemented in the virtual world Second Life, during three academic years. The students made oral presentations on topics related to the course to their peers to train their public speaking skills. The perception of the students of the new technology was generally positive, although there were some interesting differences between the different cohorts, which are discussed in detail below.

### 4.1. Training in Oral Communication Skills in Public

Communication skills should be a fundamental pillar of undergraduate training because future graduates, in addition to having sufficient knowledge of the subject, must be able to transmit these results effectively. In addition to proper written communication, oral communication is essential for health science professionals as health professionals must speak in public in a variety of settings, from formal academic meetings to informal briefings or impromptu discussions. The essential communication skills needed for these situations are the same and must be learned, practiced, and honed [23] beginning in the undergraduate years as they help enhance careers and prepare students for the job market [24]. The presenter’s voice is literally the instrument of connection with the attendees and influences how they perceive the presentation [23].

A health professional must know how to modulate his or her communication according to whom he or she is addressing; for example, medical students tend to respond more emotionally to emotional queries from patients than to their scientific queries [25]. It has been proposed as a learning objective in communication training for doctors to recognize communication styles considering the needs of patients in certain situations [25]. Interprofessional communication and collaboration are also part of the current challenges in the training of health sciences professionals since communication is the basis of interprofessional collaboration [26]. It requires specific training, and currently the use of technology is essential for this teaching task.

The application of interactive technologies in health science education requires careful selection and strategic practical implementation of e-learning tools to achieve certain curricular objectives, such as learning communication skills [27]. Technologies such as digital learning platforms are suitable for teaching interprofessional skills since they allow social and professional exchange between students of different professions [26]. Platforms such as Zoom (Zoom Video Communications, San Jose, CA, USA), Google Meet (Google, Mountain View, CA, USA), Facebook Live (Meta for Media, Menlo Park, CA, USA), Skype (Skype; Microsoft, Redmond, WA, USA), or Microsoft Teams (Microsoft Corporation, Redmond, WA, USA) enable educational experiences with synchronous communication between teacher and audience in a two-dimensional environment [28,29,30,31]. They have had great technological development since the COVID-19 pandemic and provide the advantage of allowing students to train their communication skills in public [32]. Three-dimensional environments, such as virtual worlds, also allow training communication skills in public and have some advantages over synchronous two-dimensional communication platforms [33]: (1) they induce a strong sense of presence (feeling of “being there”) in the users; (2) they promote social awareness or the ability to feel the presence and location of the participants in a learning environment, reinforcing the perception of “Who is there” and “What is happening”; and (3) they generate a greater sense of belonging to a community.

The experiences of this study focused on scientific communication skills training and peer teaching, using the immersive communication features of Second Life. Scientific communication, both written and oral, is the key to success in biomedical research, but formal instruction is rarely provided [34]. Mastering the skills to orally communicate scientific information in public requires study and practice. However, traditional undergraduate curricula often lack a focus on students’ ability to communicate in public, whether the audience is specialized or general. It is not uncommon for the student to have never presented a scientific presentation in public until his or her final degree project. Peer teaching (defined as a student teaching another student) offers many benefits, including introducing students to topics that may not be well covered in the curriculum, developing expository skills, and generating knowledge and confidence in students [35]. Peer teaching, when used with other teaching methods, offers added value to promote cooperation and social interaction among students [36], two characteristics of the educational value of virtual worlds such as Second Life [5].

Students often express different levels of self-confidence and even considerable distress when presenting scientific communications in public [34]. Although Second Life eliminates non-verbal communication, since the attendant only perceives the voice information and the slideshow along with limited gestures of the avatar, the use of avatars in the virtual environment is an advantage to break barriers in oral communication and make the student feel more comfortable speaking in public [4,37]. In this study, students delivered their presentations gaining proficiency and self-confidence in oral communication, with less shyness or embarrassment to speak in public, as expressed in open comments such as the following.


*“... It is also a good way to encourage the quietest and most timid to speak in public and express themselves since they have the advantage of anonymity and not see faces.”*


Half of the topics chosen by the students in 2016 were related to radiation risk and security measures and the other half to technological advances in medical imaging (Table 3). In 2017, the same topics were maintained to compare both cohorts, adding two about the detection of Alzheimer’s disease and the cost of radiological exams. The interest of students in these topics shows concern for the practical application and development of medical imaging and social commitment. The exposed subjects were valued as interesting and adequate for their education. Peer assessment is becoming a popular strategy for evaluating open assignments and for breaking the social isolation surrounding distance education [38]. This study provides students with training in the habit of peer evaluation, which was independent of the chosen topic, showing differences in 2016 and 2017, although some students found it subjective, recommending that this task be performed by the teacher.

Unlike solutions such as 2D teleconferencing platforms, virtual worlds such as Second Life have a greater sense of presence, providing the student the feeling of being in a certain setting [39]. In the case of our study, this was achieved through an auditorium where the student, represented by his avatar, plays a role, either in the audience or as a speaker. The communication features in Second Life simulate real-world communication, for example, sounds get louder as the avatar moves towards the source, and the volume of the voice depends on how close the users are to each other [40].

### 4.2. Students’ Perception of Experiences in Second Life

Second Life is a good tool to develop effective oral communication skills [41]. The students’ assessment of these experiences in Second Life was very positive, highlighting among all the evaluation of the teacher. This result, consistent with other previous studies involving medical students [9,10,11], reflects the students’ recognition of the teacher’s work and dedication in providing extracurricular educational activities involving new communication technologies. The students also valued the educational content and the experience very positively, providing 53% positive codes in the comments highlighting the interest, educational, and innovative value of the experience (Table 1) in comments such as the following one.


*“... It seems to me a great way to interact with both the teacher and the classmates, it is also very comfortable since it can be done from home. I hope this is not my last Second Life experience.”*


Van Ginkel et al. [42] conducted a study in which 36 first-year college students completed a compulsory oral presentation course in a virtual environment built with Unity (Unity Technologies, San Francisco, CA, USA). The participating students found the course important and motivating for developing presentation skills, with values of 4.61 ± 0.55 and 4.25 ± 0.65, respectively, on a Likert scale of 1–5. We agree with the authors that this type of experience, which imitates real life, helps to further develop oral presentation skills. The interaction with a virtual audience and the tension produced by appearing before one must be further developed in three-dimensional virtual technology. Tasks based on virtual environments should be integrated into educational practice, along with face-to-face learning processes, to practice public presentations and foster oral presentation competence [42].

In this study, the activity was voluntary in 2015 and mandatory in the following two years. It is noteworthy that, in some respects, student opinion in 2017 decreased compared to previous years. The students were more dissatisfied with the activity, the training session presenting Web pages seemed a worse idea, they found the content less suitable for their training, and they had more technical problems, with twice the number of comments on the matter compared to 2016, such as the following one.


*“…it is a very heavy program, needs a good Internet connection and uses too many processes. In my case, this has resulted in black screen sessions because my computer couldn’t handle it, which is not good.”*


The technology acceptance model developed for the acceptance of three-dimensional virtual worlds [43] includes interrelated variables such as visual appeal, perceived ease of use, perceived usefulness, perceived enjoyment, and computing self-efficacy. The lower acceptance of virtual world technology in 2017 may be related to technical aspects such as the lack of access to Second Life from the School’s computers or the lack of bandwidth of the institutional Wi-Fi, gut it can also be related to a counterproductive effect of a compulsory activity, described as “mandatory fun” in the work environment [44] and also seen in radiology educational games in Second Life, in which compulsory participation decreased the acceptance of the technology of the virtual world and the opinion about the experience [20]. The compulsory activities are part of the extrinsic motivation of the students, but in the case of “imposed” fun or entertaining activities such as those in Second Life, it is possible that the students do not find it rewarding enough considering the cognitive effort that may be involved in performing activities in the virtual world.

As in the present study, in general, university students and clinical professors describe positive educational experiences with Second Life, which suggests the need for further research on its application in university education [45]. From the point of view of the teachers who participated in in this study, the learning and training of public speaking skills should be further studied in Second Life. The simulation of being in front of a virtual audience should be further explored, and it will be interesting to compare the perception of students from different degree programs, for example, biomedical engineering and medicine.

### 4.3. Limitations and Future Perspectives

Among the limitations of the present study, it is worth highlighting those related to technology. Access to and proper representation of the virtual world requires sufficient computer graphics capacity and adequate bandwidth to transmit data over the Internet. Problems caused by technical limitations can vary in degree of user rejection, so the experience in the virtual world was not consistent for all participants [5]. Given that this study was conducted between 2015 and 2017, the rapid technological advancement in computing solutions may suggest that there are fewer technological challenges to using platforms such as Second Life today and fewer in the foreseeable future. It is essential that students learn to use audio communication well in Second Life (the microphone and headphones) because the lack of fluency in communication slows down the development of a session, annoying others and decreasing their attention, as pointed out by several students of the group 2017.

Since there is no direct eye contact, it is somewhat more difficult to keep the student motivated during the session and to prevent the audience from adopting a passive attitude. Conducting synchronous sessions in Second Life exclusively with the voice and getting students to maintain the appropriate degree of attention and participation requires a certain training, so current experience should be used for similar future projects.

The implementation of these learning tools represents a significant investment of time by teachers in the design, development, and management of these virtual spaces. Technological resources and pedagogical designs that work must be reused to make the time invested profitable. Considering the results of this study, it is interesting to delve into the training activities of students as speakers. This type of activity was developed in the following courses (2018 and 2019), with bioengineering and medicine students, to compare the perception of both cohorts of students. Other possibilities of Second Life in biomedical engineering education include holding conferences with invited professors, reducing travel costs [46], conducting competitive learning games individually [19] or in teams [47], or implementing virtual laboratories [6].

This study dates back a few years, and since 2020 the COVID-19 pandemic has changed and continues to change the way of approaching higher education [18], with a large increase in the use of synchronous communication technologies providing a new horizon for teaching in virtual worlds, with acceptance from students and teachers [48].

## 5. Conclusions

This study provides the perception of biomedical engineering students in various online learning experiences on medical imaging complementary to the formal course in an immersive, novel, and playful environment, Second Life.

In general, the students’ assessment of these experiences in Second Life was very positive, highlighting the teacher, the island environment, and the educational content. They found the experiences interesting, educational, and innovative. The students valued Second Life as an attractive and suitable environment for their training in science communication skills. Second Life enables students to present scientific content effectively to their peers, receiving hands-on training in the tasks of collecting, organizing, and presenting data and with the benefits of remote access, collaborative work, and social interaction. Being a non-face-to-face activity, it constitutes a good training option since students gain self-confidence and are less afraid of speaking in public.

## Figures and Tables

**Figure 1 ijerph-20-01697-f001:**
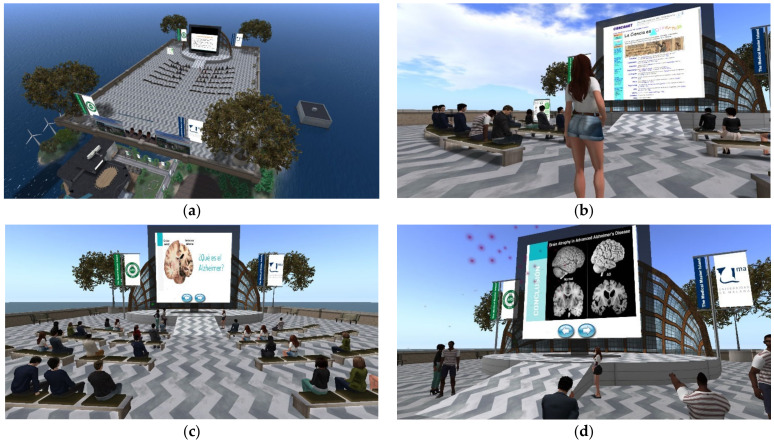
(**a**) Screenshot showing an aerial view of the floating auditorium located 120 m above the ground where the activities of this study were carried out. (**b**) Scene with a female student showing her classmates a web page displayed on the presentation screen. (**c**,**d**) Scenes with a group of students in front of a panel showing the slide presentation on neuroimaging in Alzheimer’s disease. The forward and backward buttons allow navigating the web-based slideshow.

**Figure 2 ijerph-20-01697-f002:**
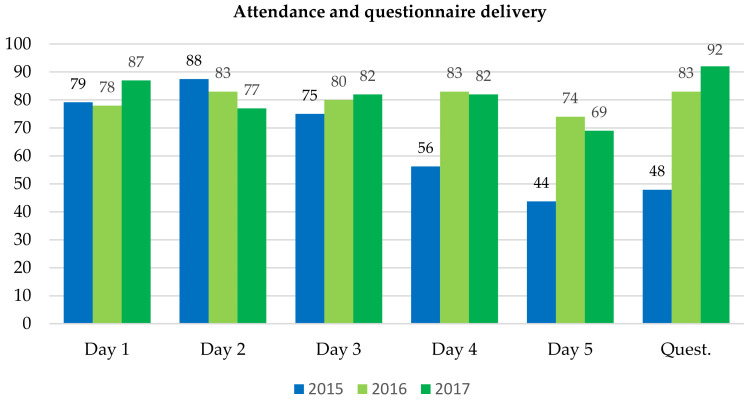
Percentage of attendance to online activities and questionnaire delivery (Quest) in the experiences included in this study: a pilot experience with students as presenters in 2015 (N = 48) and training of students as presenters of oral communications in 2016 (N = 46) and 2017 (N = 39).

**Figure 3 ijerph-20-01697-f003:**
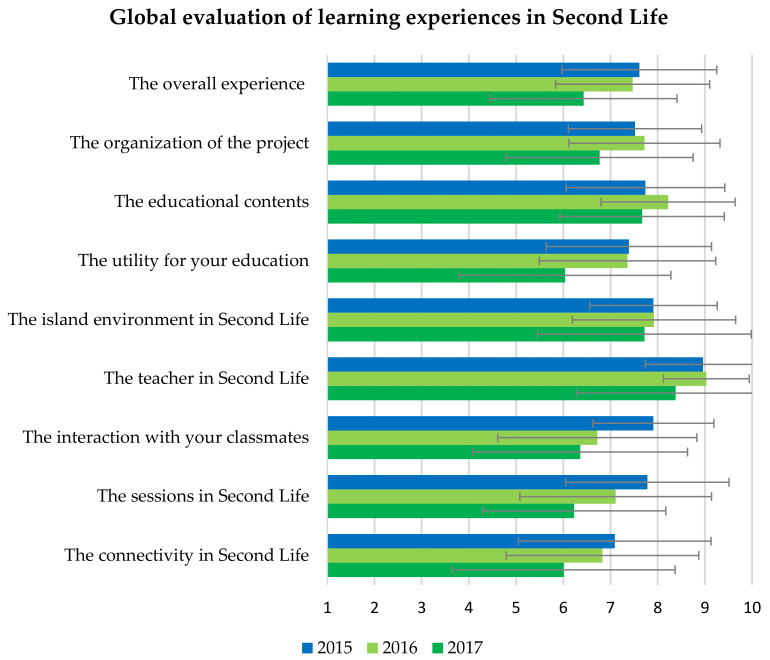
Comparison of the global evaluation, from 1 to 10 points, on the experiences in Second Life included in this study. The results represent the mean of each group. Error bars represent the standard deviation.

**Figure 4 ijerph-20-01697-f004:**
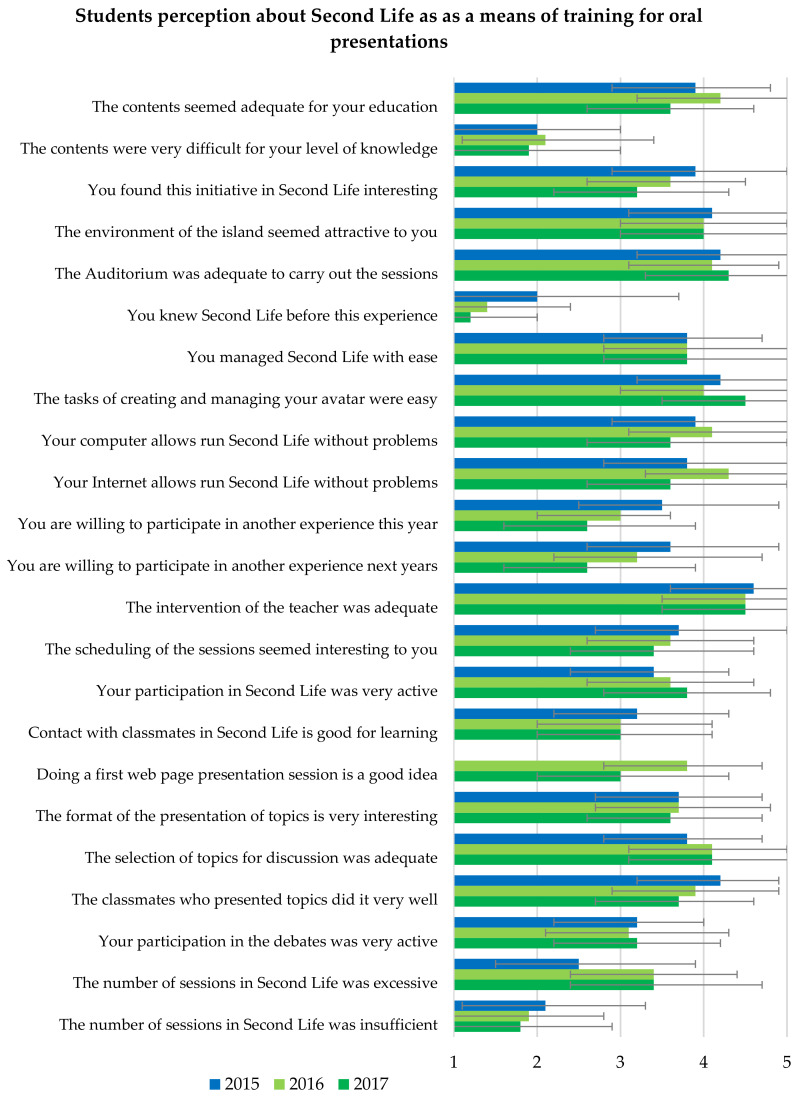
Bar diagram comparing the responses on a Likert scale from 1 (completely disagree) to 5 (completely agree) for the statements about the experiences of the students presenting oral expositions to their classmates. The data represent the mean and the error bars the standard deviation.

**Table 1 ijerph-20-01697-t001:** Coding of open comments provided in the questionnaires.

Codes	2015	2016	2017	Total
**Positive**	**10**	**36**	**26**	**72**
Interesting	1	11	8	**20**
Educational	4	10	7	**21**
Innovative	2	7	5	**14**
Ubiquity	-	2	3	**5**
Fun	2	1	2	**5**
Willingness	1	2	-	**3**
Gratitude	-	3	1	**4**
**Negative**	**3**	**24**	**26**	**53**
Technical	1	5	10	**16**
Schedule	-	4	3	**7**
Distraction	-	4	-	**4**
Time	2	3	7	**12**
Face-to-face better	-	4	3	**7**
School computers	-	-	2	**2**
Assessment	-	4	1	**5**
**Suggestion**	**1**	**6**	**4**	**11**
**Total**	**14**	**66**	**56**	**136**

**Table 2 ijerph-20-01697-t002:** Score given by the students to the oral presentations presented on the fifth day in the 2015 experience.

Title of the Presentation	First ^1^	Second ^1^	Third ^1^	Score	Normalized
Orthopantomography	9	8	0	13.0	6.2
Tomosynthesis: 3D mammography	5	4	6	8.8	4.2
Elastography	4	5	2	7.1	3.4
RM spectroscopy	0	4	9	4.7	2.2
Contrast-enhanced spectral mammography	3	0	3	3.9	1.9

^1^ The results are the number of times each presentation was rated by classmates as the best or second or third best presentation. Score is the sum of points, awarding 1.0, 0.5, or 0.3 depending on whether they were considered the best or second or third best presentation. The last column is the normalized points over 10.

**Table 3 ijerph-20-01697-t003:** Peer evaluation of oral presentations given in 2016 and 2017.

Oral Presentation Title (Topic)	Year	N	Contents	Quality	Understand. ^1^	Interest
Risks of radiological studies in childhood	2016	3	8.0 ± 1.2	8.2 ± 1.2	8.5 ± 1.1	8.6 ± 1.2
	2017	3	8.3 ± 1.2	8.1 ± 1.2	8.0 ± 1.3	8.4 ± 1.0
How many X-rays is it safe to have per year?	2016	4	8.4 ± 1.0	8.3 ± 0.9	8.4 ± 1.0	8.5 ± 1.1
	2017	2	7.5 ± 1.2	7.0 ± 1.3	7.5 ± 1.1	8.5 ± 1.2
Risks of X-rays in pregnant women	2016	3	8.6 ± 1.0	8.5 ± 1.1	8.4 ± 1.1	8.8 ± 1.2
	2017	3	8.2 ± 1.2	7.9 ± 1.0	8.3 ± 1.2	8.8 ± 1.1
Security measures in areas where X-rays are used	2016	4	8.4 ± 0.8	8.3 ± 1.2	8.6 ± 1.0	8.5 ± 1.3
	2017	3	7.9 ± 1.9	8.3 ± 1.4	8.6 ± 1.2	8.6 ± 1.2
How radiation affects people according to age	2016	3	8.7 ± 1.0	8.9 ± 0.9	8.9 ± 1.0	9.1 ± 0.9
	2017	3	7.7 ± 1.3	7.7 ± 1.3	8.0 ± 1.2	8.1 ± 2.0
Security measures in areas where radioactive isotopes are used	2016	2	8.3 ± 1.0	8.4 ± 1.1	8.3 ± 1.1	8.6 ± 1.1
	2017	3	8.0 ± 1.9	8.1 ± 1.7	8.2 ± 1.6	8.4 ± 1.6
Contrast-enhanced spectral mammography	2016	4	8.9 ± 0.8	8.7 ± 1.1	8.5 ± 0.9	8.9 ± 1.0
	2017	3	7.8 ± 1.1	6.9 ± 1.3	7.3 ± 1.2	7.6 ± 1.3
What is the diffusion tensor and MR tractography?	2016	4	8.5 ± 1.1	8.4 ± 1.2	8.2 ± 1.1	8.6 ± 1.2
	2017	3	8.0 ± 1.3	7.2 ± 1.5	7.1 ± 1.4	7.6 ± 1.5
What is tomosynthesis and what is it for?	2016	4	8.3 ± 1.0	8.3 ± 1.1	8.1 ± 1.4	8.5 ± 1.1
	2017	3	8.1 ± 1.1	8.1 ± 1.0	8.2 ± 0.9	8.4 ± 1.3
What is MR spectroscopy and what is it for?	2016	4	8.6 ± 1.2	7.9 ± 2.0	7.9 ± 1.3	8.4 ± 1.3
	2017	3	8.0 ± 1.1	7.9 ± 1.1	7.8 ± 1.3	7.9 ± 1.5
What is elastography and what is it for?	2016	3	8.4 ± 1.0	8.5 ± 1.3	8.3 ± 1.3	8.3 ± 1.3
	2017	2	7.6 ± 1.0	7.5 ± 1.2	7.5 ± 1.5	7.9 ± 1.4
What is functional MRI and what is it for?	2016	4	8.1 ± 1.1	8.2 ± 1.2	8.0 ± 1.0	8.6 ± 1.1
	2017	2	5.8 ± 1.6	6.2 ± 1.0	6.1 ± 1.4	7.3 ± 1.3
Can Alzheimer’s be diagnosed with biomedical imaging?	2016	-	-	-	-	-
	2017	3	8.2 ± 1.2	8.2 ± 1.1	7.6 ± 1.3	8.5 ± 1.3
How much do radiological studies cost? Can costs be reduced?	2016	-	-	-	-	-
	2017	2	8.7 ± 0.9	8.5 ± 0.9	8.8 ± 0.7	8.7 ± 0.9
**All the presentations**	**2016**	**-**	**8.4 ± 1.0**	**8.4 ± 1.3**	**8.3 ± 1.1**	**8.6 ± 1.2**
	**2017**	**-**	**7.9 ± 1.4**	**7.7 ± 1.4**	**7.8 ± 1.4**	**8.2 ± 1.4**

The values represent the mean ± standard deviation of the scores (from 1 to 10) given by classmates for the content and quality of the presentation and the understandability of and interest in the topic. N is the number of students presenting the oral presentation. ^1^ Understandability.

## Data Availability

The datasets used and/or analyzed during the current study are available from the corresponding author on reasonable request.

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
