# Peer review of "Training Scientific Communication Skills on Medical Imaging within the Virtual World Second Life: Perception of Biomedical Engineering Students"

_ijerph, 2023, doi:10.3390/ijerph20031697_

Round 1
Reviewer 1 Report (New Reviewer)
The article is well written, the results are properly analyzed, and its content is relevant, especially in the context of the changes in teaching we have experienced due to the known pandemic.
In my opinion, the paper can be published almost as it is. What I missed was a comparison of the results of surveys and students' impressions with the results of similar analyzes present in the literature. I would suggest expanding the discussion with such a comparison.
Perhaps it would also be worth writing a bit about what the lecturers think about the results of the surveys, as well as to describe their impressions.
Author Response
The article is well written, the results are properly analyzed, and its content is relevant, especially in the context of the changes in teaching we have experienced due to the known pandemic.
Response: Thank you very much for those comments about our article, they are very stimulating for us.
In my opinion, the paper can be published almost as it is. What I missed was a comparison of the results of surveys and students' impressions with the results of similar analyzes present in the literature. I would suggest expanding the discussion with such a comparison.
Response: Thank you very much for this comment. As already expressed in the manuscript, the opinions of the students overlap with those of other studies previously carried out with medical students [references 9-10 and 11]. We have included a paragraph referring to the interesting study by van Ginkel et al., in which undergraduate students find the experience in virtual worlds useful and motivating to improve oral communication skills in public. Literally "We agree with the authors that this type of experience, which imitates real life, helps to further develop oral presentation skills" and therefore, it is necessary to continue deepening the application of this technology.
Perhaps it would also be worth writing a bit about what the lecturers think about the results of the surveys, as well as to describe their impressions.
Response: Thank you very much for the comment. Actually, the opinion of the lecturers, as part of the authors, is expressed throughout the discussion. In addition, the following paragraph has been included: "As in the present study, in general, university students and clinical professors describe positive educational experiences with Second Life, which suggests the need for further research on its application in university education [MauldinPereira-2018] From the point of view of the teachers who participated in this study, the learning and training of public speaking skills should be further studied in Second Life. The simulation of being in front of a virtual audience should be explored, and it will be interesting to compare the perception of students from different degrees, for example biomedical engineering and medicine".

Reviewer 2 Report (New Reviewer)
This is a well written report of a well conducted single center observational study that accompanied the undergraduate training of third year biomedical engineering students concerning biomedical imaging. Subject of the observation was the attempt to teach biomedical imaging via immersion in a virtual world, using Second Life. Data source is voluntarily disclosed information by the students via questionnaires, which were well analysed.
The authors describe that the perception of the students of the new technology was generally positive and that they s valued Second Life as an attractive and suitable environment for their training in science communication skills. They conclude that Second Life constitutes a good training option, since students gain selfconfidence and are less afraid of speaking in public.
Author Response
Response: Thank you very much for your words about our paper and for the synthesis you make of the work done. We are truly excited to be able to publish it.

This manuscript is a resubmission of an earlier submission. The following is a list of the peer review reports and author responses from that submission.
Round 1
Reviewer 1 Report
The Authors address a very novel, and current topic of virtual reality and its applications in university and professional activities. Although the implementation of virtual metaverse, i.e. Second Life seems to be an attractive solution, especially in times of pandemia, there are several concerns regarding the study design: (1) the study comprises four pilot sections - each including a small number of about 40-45 participants with the questionnaire response rate of 48 to 92%, leaving a very small data set for further statistical analysis; (2) the delivered questionnaires assessed in Likert scale formulate very general questions regarding the overall using experience; to reviewer's opinion questions should be focused on rating more particular features of the software and virtual tools provided by Second Life, and its efficiency in delivering academic presentations, form communities, etc.; (3) the majority of relevant data regarding users' impressions is delivered through unstructured open comments; a substantial number of subjects pointed to the technical concerns and system overload with Second Life - as the study was conducted between 2014 and 2017, the rapid technological progress in computer solutions make data a bit outdated. Other concerns included class schedule, and software availability (not provided by the University) - which are not exactly the focus of the paper; (4) the study (design of questionnaires) unfortunately does not allow participants to compare their experience with real-life learning to the virtual one, so an interesting set of data is missed.
All in all, to reviewer's opinion, although interesting in nature, the manuscript is lacking some major methodological points to be considered for publication in IJERPH.
Author Response
Reviewer 1
The Authors address a very novel, and current topic of virtual reality and its applications in university and professional activities. Although the implementation of virtual metaverse, i.e. Second Life seems to be an attractive solution, especially in times of pandemia, there are several concerns regarding the study design:
R: We thank reviewer 1 for his/her opinion on the implementation of Second Life technology in our study on radiology education for biomedical engineering students. Thank you for providing considerations and concerns about our manuscript. We answer them in detail below, hopefully correctly. Please note that in the current version of the manuscript the 2014 experience has been removed, based on reviewer 2's last comment, since the activity developed in Second Life (attending virtual classes on 2D and 3D platforms and then comparing them) is well different from the years 2015-2017.
(1) the study comprises four pilot sections - each including a small number of about 40-45 participants with the questionnaire response rate of 48 to 92%, leaving a very small data set for further statistical analysis;
R: Thank you for this comment. The study, now limited to 3 years but with a common methodology (training the oral presentation in public of the students), intends to make the reader aware of the implementation of this technology as an additional contribution to the program of a subject on radiological images in biomedical engineering and to evaluate the perception of the students. Presenting a pilot and reproducing the same educational strategy for two more courses. With all due respect, we do not consider the current sample of 133 students and 97 submitted questionnaires to be too small to draw any conclusions.
(2) the delivered questionnaires assessed in Likert scale formulate very general questions regarding the overall using experience; to reviewer's opinion questions should be focused on rating more particular features of the software and virtual tools provided by Second Life, and its efficiency in delivering academic presentations, form communities, etc.;
R: Thank you for this comment. The general questions used in the questionnaires allow comparing the opinion of different positions in various experiences. The questionnaires used in this study have been used in other published studies on teaching experiences with medical students (new reference numbers 9–11,19,20), with some minimal modification adapting the statements to the current study. The questionnaire showed good reliability for the statements related to the teaching activity by the students and the global evaluation of the project (Cronbach's alpha ≥0.84) and reliability of the statements about the experience of the students in Second Acceptable Life (Cronbach's alpha ≥0.70). Kendall's Tau-b analysis showed a positive correlation between all paired statement comparisons related to global assessment (Tau-b > 0.22, P < 0.01), and students' perception of the experience (Tau-b > 0.15, P < 0.05) (new reference numbers 19,20). This information has been included in the methods of the current version, the first time the questionnaire is described (page 4 line 162 and following).
We want to acknowledge the reviewer's suggestion to incorporate more specific questions in future projects.
(3) the majority of relevant data regarding users' impressions is delivered through unstructured open comments; a substantial number of subjects pointed to the technical concerns and system overload with Second Life - as the study was conducted between 2014 and 2017, the rapid technological progress in computer solutions make data a bit outdated. Other concerns included class schedule, and software availability (not provided by the University) - which are not exactly the focus of the paper;
R: Thank you for this observation. We have included the following in the Limitations and future perspectives section (page 13, line 422): “Given that this study was conducted between 2015 and 2017, the rapid technological advancement in computing solutions may suggest that there are fewer technological challenges to using platforms like Second Life today and fewer in the foreseeable future.”
(4) the study (design of questionnaires) unfortunately does not allow participants to compare their experience with real-life learning to the virtual one, so an interesting set of data is missed.
R: Thank you for this comment, but the objective of this study is not to compare with real life, but to provide the students' perception of teaching activities in 3D worlds, complementary to regulated education. The comparison with the real world has been made in other previous studies, such as (new reference number 11):
Lorenzo-Alvarez, R.; Tudolphi-Solero, T.; Ruiz-Gomez, M.J.; Sendra-Portero, F. Medical student education for abdominal radiographs in a 3D virtual classroom versus traditional classroom: A randomized controlled trial. AJR Am J Roentgenol 2019, 213, 644–650. [doi.org/10.2214/AJR.19.21131]
All in all, to reviewer's opinion, although interesting in nature, the manuscript is lacking some major methodological points to be considered for publication in IJERPH.
R: Thank you for your comments and contributions. We hope that we have clarified these methodological points with the responses to both reviewer 1 and 2, and that the current version qualifies for consideration for publication in this special issue of IJERPH.
Reviewer 2 Report
Regarding the introduction and background, I felt the authors could have better organized the content. My suggestion would include discussing Second Life in the introduction earlier to provide context for the purpose of the study. Perhaps, after discussing the Second Life platform, the authors could then provide support for the need for virtual technology in medical imaging teaching experiences. Overall, the introduction did not represent the article well as it seemed unorganized and disjointed. The purpose of the article as written in the objectives was vague but appeared to be sufficient. Regarding materials and methods, the section headings attempt to describe the research process in succession, from design to course description, pilot testing and the research experiment. However, the methods described were not chronological. The data analysis portion of the article was suffice and the results were clearly presented. The conclusion provided more insight into the research design than other areas of the article. Overall, the article needs to be better organized and the statement and purposed needs to be more clearly stated. The research is not relevant or useful to the teaching environment based on the virtual reality platform utilized.
Ultimately, MRI/CT/x-ray images shown in Second Life loose quality because they are seen from far in the screen. Second Life should not have been used for these specific courses. Second Life could have been considered as a platform of choice if authors where trying to investigate interaction among students.
Collecting data over the years is difficult in research. Authors tried to compare years however they have too much variation: type of content, number of students, length of program. I recommend presenting this research at a conference.
Author Response
Reviewer 2
Regarding the introduction and background, I felt the authors could have better organized the content. My suggestion would include discussing Second Life in the introduction earlier to provide context for the purpose of the study. Perhaps, after discussing the Second Life platform, the authors could then provide support for the need for virtual technology in medical imaging teaching experiences.
R: Thank you for your comment on the introduction and background. We have modified it accordingly, like this: 1.1 Virtual worlds in health-related undergraduate education; 1.2 Teaching of radiology to Biomedical Engineering students; 1.3 Background; 1.4 Objectives
Overall, the introduction did not represent the article well as it seemed unorganized and disjointed. The purpose of the article as written in the objectives was vague but appeared to be sufficient.
R: Thank you for this comment. As we have indicated in the previous point, the focus of the introduction has been changed, placing more emphasis on the need for virtual technology for teaching radiology.
In the purpose (objectives) we have eliminated the experience of 2014 (according to your last comment) to give more homogeneity to the educational method, now focused exclusively on training in oral presentations as part of the students’ education.
Regarding materials and methods, the section headings attempt to describe the research process in succession, from design to course description, pilot testing and the research experiment. However, the methods described were not chronological.
R: Thanks. The study seeks for the reader to understand what teaching activity has been carried out in the three years and in what context, therefore: 1) the virtual environment is briefly described; 2) the Biomedical Imaging course is contextualized; and 3) the three experiences, one pilot and the following two, are described.
The data analysis portion of the article was suffice and the results were clearly presented.
R: Thanks for this comment.
The conclusion provided more insight into the research design than other areas of the article.
R: Thank you for this comment. We have deleted the first sentence of the conclusions "Efforts by biomedical engineering programs to provide robust education in biomedical imaging have far-reaching benefits and should be encouraged [2]. " We have excluded the conclusion relative to the 2014 experience. The rest summarizes, in our opinion, the most relevant aspects of the study.
Overall, the article needs to be better organized and the statement and purposed needs to be more clearly stated. The research is not relevant or useful to the teaching environment based on the virtual reality platform utilized.
R: Thank you, in response to comments from reviewers 1 and 2 the contents have been modified and reorganized. As stated at the beginning of the discussion, innovative technological efforts are needed to engage current biomedical engineering students in biomedical imaging, and 3D virtual worlds such as Second Life constitute a promising innovation, with unique characteristics that can have a positive impact on teaching and learning, such as: immersion, sense of presence, visualization, contextualization, content simulation and individual or collective interactions.
Ultimately, MRI/CT/x-ray images shown in Second Life loose quality because they are seen from far in the screen. Second Life should not have been used for these specific courses.
R: A: Thank you for that comment. The images in figure 1 are illustrative, so that the reader understands the aspect of the virtual world and the activities carried out in it. Radiological images and adapted PowerPoint presentations are clear enough to be displayed in Second Life, as the user can modify his point of view (the avatar camera) appropriately. In fact, it is one of the aspects that is trained specifically on the first day of each activity.
Second Life could have been considered as a platform of choice if authors where trying to investigate interaction among students.
R: We appreciate this comment. Interaction in the metaverse in any of its modalities is essential. The user interacts with the environment and that is the basis of his/her performance in it. In multi-user virtual worlds like Second Life, interaction with other users is a great advantage. Certainly the interaction between students is an added value of this experience (it is specifically asked about in the questionnaire), but it is something that should be sought in any current higher education context. That students interact with each other, virtual or face to face, is an objective that every teacher should set, as it enriches learning.
Collecting data over the years is difficult in research. Authors tried to compare years however they have too much variation: type of content, number of students, length of program. I recommend presenting this research at a conference.
R: Thank you for this comment which leads to the main modification of the manuscript. It was originally intended to chronologically describe our approach to teaching Second Life with biomedical engineering students. But, comparing the experience of 2014, with an educational design and structure very different from other years, represents a great variation and blurs the study. For this reason, it has been eliminated, leaving the experiences of the years 2015 to 2017, focused on training students in oral presentation skills, with topics that they themselves propose, related to the syllabus of the subject. In our opinion, the current structure is more coherent, a first year of piloting and two years of reproducing and improving the experience.
Round 2
Reviewer 1 Report
-
Reviewer 2 Report
Thank you for working on an improved revised version.Removing the data related to the 2014 experience was a good idea. However, there are still several issues:
1) using Second Life to teach or to allow students to give presentation is not the main goal of Second Life. You can use powerpoint, zoom, or any other teleconferencing platforms. Using such platforms would also allow for a better quality of the images shown. It would have been interesting to use Second Life for a group project and to study how students interacted. Interaction is the main goal of Second Life.
2) The qualitative data in the discussion are presented without being processed using a common methodology for analyzing qualitative data.
3) In order to truly assess the effectiveness of using Second Life to teach a concept, authors should have used a knowledge assessment such as a quiz or exam.
I think the data available are not enough to make a case for a journal article